

# Antimicrobial stewardship in remote primary healthcare across northern Australia

Will Cuningham[1], Lorraine Anderson[2], Asha C. Bowen[1,3,4],
Kirsty Buising[5,6], Christine Connors[7], Kathryn Daveson[8,9],
Joanna Martin[2], Stacey McNamara[8], Bhavini Patel[7,10], Rodney James[5],
John Shanks[7], Kerr Wright[2], Trent Yarwood[8,11,12,13],
Steven YC Tong[1,14] and Jodie McVernon[5,15]

[1] Menzies School of Health Research, Charles Darwin University, Darwin, Northern Territory, Australia
[2] Kimberley Aboriginal Medical Services, Kimberley, Western Australia, Australia
[3] Wesfarmers Centre for Vaccines and Infectious Diseases, Telethon Kids Institute, University of Western Australia, Perth, Western Australia, Australia
[4] Department of Infectious Diseases, Perth Children's Hospital, Perth, Western Australia, Australia
[5] The Peter Doherty Institute for Infection and Immunity, The Royal Melbourne Hospital and The University of Melbourne, Melbourne, Victoria, Australia
[6] Victorian Infectious Diseases Service, The Royal Melbourne Hospital, Melbourne, Victoria, Australia
[7] Top End Health Service, NT Department of Health, Darwin, Northern Territory, Australia
[8] Queensland Statewide Antimicrobial Stewardship Program, Metro North Hospital and Health Service, Brisbane, Queensland, Australia
[9] Department of Infectious Diseases and Microbiology, Canberra Hospital, Canberra, Australian Capital Territory, Australia
[10] Charles Darwin University, Darwin, Northern Territory, Australia
[11] Cairns Hospital, Cairns, Queensland, Australia
[12] Rural Clinical School, University of Queensland, Brisbane, Queensland, Australia
[13] College of Medicine and Dentistry, James Cook University, Townsville, Queensland, Australia
[14] Victorian Infectious Diseases Service, The Royal Melbourne Hospital, and Doherty Department University of Melbourne, at the Peter Doherty Institute for Infection and Immunity, Melbourne, Victoria, Australia
[15] Melbourne School of Population and Global Health, The University of Melbourne, Melbourne, Victoria, Australia

Corresponding author
Jodie McVernon,
j.mcvernon@unimelb.edu.au

## ABSTRACT

**Background:** The high burden of infectious disease and associated antimicrobial use likely contribute to the emergence of antimicrobial resistance in remote Australian Aboriginal communities. We aimed to develop and apply context-specific tools to audit antimicrobial use in the remote primary healthcare setting.

**Methods:** We adapted the General Practice version of the National Antimicrobial Prescribing Survey (GP NAPS) tool to audit antimicrobial use over 2–3 weeks in 15 remote primary healthcare clinics across the Kimberley region of Western Australia (03/2018–06/2018), Top End of the Northern Territory (08/2017–09/2017) and far north Queensland (05/2018–06/2018). At each clinic we reviewed consecutive clinic presentations until 30 presentations where antimicrobials had been used were included in the audit. Data recorded included the antimicrobials used, indications and treating health professional. We assessed the appropriateness of antimicrobial use and functionality of the tool.

**Results:** We audited the use of 668 antimicrobials. Skin and soft tissue infections were the dominant treatment indications (WA: 35%; NT: 29%; QLD: 40%). Compared with other settings in Australia, narrow spectrum antimicrobials like benzathine benzylpenicillin were commonly given and the appropriateness of use was high (WA: 91%; NT: 82%; QLD: 65%). While the audit was informative, non-integration with practice software made the process manually intensive.
**Conclusions:** Patterns of antimicrobial use in remote primary care are different from other settings in Australia. The adapted GP NAPS tool functioned well in this pilot study and has the potential for integration into clinical care. Regular stewardship audits would be facilitated by improved data extraction systems.

## INTRODUCTION

Antimicrobial resistance (AMR) is a major healthcare issue. Compared with other countries, Australia currently has relatively low (albeit rising) rates of AMR, despite comparatively high rates of antimicrobial use in humans (*Cecchini, Langer & Slawomirski, 2015*; *Organisation for Economic Co-operation and Development, 2016*; *Klein et al., 2018*; *Van Boeckel et al., 2014*). Remote regions in northern Australia, however, have some of the highest rates of AMR in the world; for example, at least 40% of *Staphylococcus aureus* community-associated isolates are methicillin-resistant (MRSA) (*Australian Commission on Safety and Quality in Health Care (ACSQHC), 2019a*; *ACSQHC, 2018b*; *Tong et al., 2008*). In comparison, rates of MRSA are about 15–20% elsewhere in Australia. Recognising the threat of emerging resistance to human health, the Australian Government Departments of Health and Agriculture published the first National Antimicrobial Resistance Strategy in June 2015 (*Australian Government Department of Health and Department of Agriculture, 2015*). The accompanying implementation plan highlighted areas for specific focus including identification of barriers and enablers to antimicrobial stewardship (AMS) in Indigenous healthcare organizations to inform development of setting-specific resources (*Australian Government Department of Health and Department of Agriculture and Water Resources, 2016*). Addressing AMR in Indigenous primary healthcare and coordinating AMS programs in partnership with Aboriginal and Torres Strait Islander communities have been highlighted as priorities (*Bowen et al., 2019*).

Australia is a geographically large country. Northern Australia alone (the region north of the Tropic of Capricorn) is approximately equivalent in area to Argentina and is sparsely populated. The Northern Territory (NT), for example, has a population density of 0.2 people/km$^2$, the lowest of Australia's eight jurisdictions (*Australian Bureau of Statistics, 2016a*). Of its 245,000 residents, 30% live in rural areas (compared with 10% Australia-wide) (*Australian Bureau of Statistics, 2019*). Furthermore, while only 3% of the Australian population are Aboriginal and/or Torres Strait Islander, this is 30% in the NT (*Australian Bureau of Statistics, 2016b*). Of these Aboriginal and Torres Strait Islander
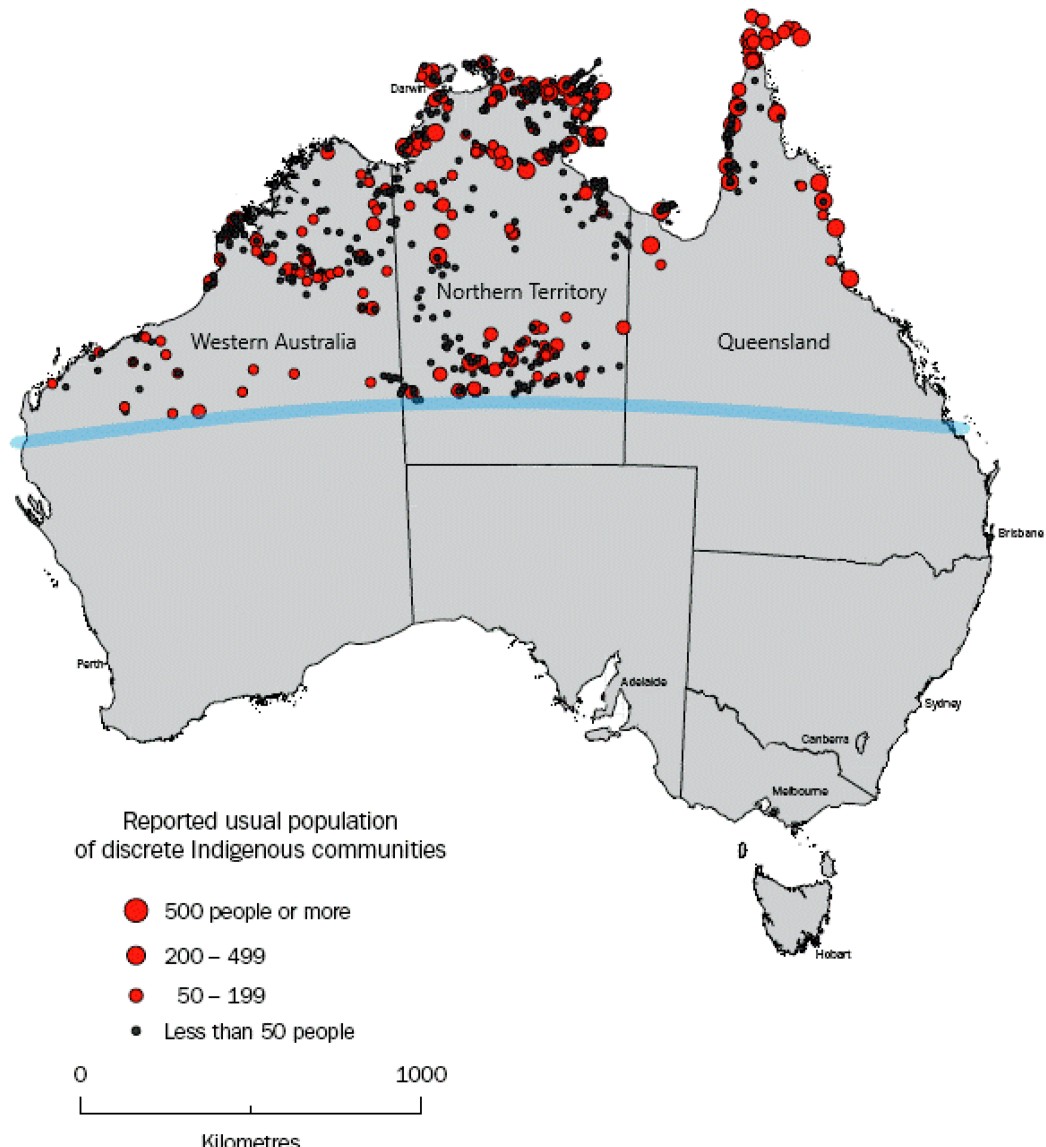

Reported usual population
of discrete Indigenous communities

● 500 people or more

● 200 – 499

● 50 – 199

• Less than 50 people

0                    1000

Kilometres

**Figure 1 Map adapted from the Australian Bureau of Statistics, showing location and population of discrete Indigenous communities in northern Australia (i.e., above the Tropic of Capricorn (blue line)).** Source: Australian Bureau of Statistics, 2006, Housing and Infrastructure in Aboriginal and Torres Strait Islander Communities, cat. no. 4710.0, viewed 5 May 2020, https://www.abs.gov.au/AUSSTATS/abs@.nsf/Latestproducts/4710.0Appendix42006?opendocument&tabname=Notes&prodno=4710.0&issue=2006&num=&view=.     

peoples, 81% live remotely (calculated using an index reflecting the community's population size and level of access to goods and services), often many hundreds, if not thousands of kilometers from the nearest urban center (Fig. 1). In northern Western Australia (WA) and Queensland (QLD), there are similar sparsely populated communities. These communities have small healthcare clinics, often Aboriginal Community Controlled Health Services and antimicrobials are routinely supplied through the Pharmaceutical Benefits Scheme (PBS). The health workforce is varied, comprised of doctors, nurses, Aboriginal Health Practitioners and other health workers, many of whom are on

temporary contracts or work on a fly-in/fly-out basis. The mean annual staff turnover rate in this setting is as high as 128% (*Russell et al., 2017*). Usually more severe cases of infection require treatment at a larger, urban hospital which involves air transport.

The high burden of infectious diseases in remote communities necessitates high antimicrobial use, emphasizing the need for good AMS (*Cuningham et al., 2019*). However, specific information about prescribing in remote settings is difficult to obtain. For example, the emergence of trimethoprim-resistant clones of community associated MRSA in the NT (*Tong et al., 2015*), WA (*Coombs, Pearson & Robinson, 2015*) and Queensland (*Guthridge et al., 2019*) may be related to recommendations for co-trimoxazole in skin sore treatment (*Bowen et al., 2014*), but this hypothesis cannot be tested due to a lack of prescribing data.

Hospital-based AMS programs are tasked with monitoring antimicrobial use and implementing AMS procedures in northern Australian hospitals (Kimberley Aboriginal Health Planning Forum (KAHPF) stewardship committee, WA; Top End Health Service (TEHS) stewardship committee, NT; Queensland Statewide Antimicrobial Stewardship Program (QSAMSP)) to meet accreditation standards (*ACSQHC, 2017a*, *2017c*). Extension of stewardship into the remote primary healthcare sector has been minimal due to a lack of related accreditation requirements and audit tools, limited workforce and competing priorities. Authorised healthcare workers other than doctors, such as registered nurses and Aboriginal Health Practitioners may supply and administer antimicrobials in remote settings, in accordance with local treatment protocols and jurisdictional legislation (*Kimberley Aboriginal Health Planning Forum (KAHPF), 2016*; *Remote Primary Health Care Manuals, 2017*; *Queensland Health, Royal Flying Doctor Service, 2019*). Context-specific tools are needed to capture prescribing practices and provide evidence-based feedback and indicators to support best clinical practice and health outcomes.

Hospital-based National Antimicrobial Prescribing Surveys (NAPS) have been conducted since 2011 and provide a yearly snapshot of antimicrobial use, indications and appropriateness in reference to local and national guidelines (*National Centre for Antimicrobial Stewardship (NCAS) and ACSQHC, 2018*). Participation in the NAPS is voluntary, 314 hospitals nation-wide participating in the latest report. Appropriateness of antimicrobial use in the hospital setting was 76%, cefazolin, ceftriaxone and amoxicillin-clavulanate being the most commonly used (*NCAS and ACSQHC, 2018*; *SA Health, ACSQHC, 2018*). Other studies have highlighted some differences in antimicrobial use and challenges to AMS in the regional and remote hospital setting (*Bishop et al., 2018*, *2019a*, *2019b*; *Broom, Broom & Kirby, 2014*). Related AMS tools are being developed to assess antimicrobial prescribing practices in urban General Practices (GP). Data on antimicrobial use in GP is limited, particularly in remote primary healthcare. Some data collected through the PBS and GP electronic software show the frequent use of penicillins with extended spectrum and β-lactamase inhibitor combinations (*ACSQHC, 2019a*). Rates of antimicrobial use have decreased in recent years, but there continues to be overuse compared with guideline recommendations and inappropriate use of antibiotics for respiratory indications like acute bronchitis and influenza (*ACSQHC, 2019a*). Crucially, antimicrobial use in the remote northern Australian setting is not captured by the PBS, leaving a gap in our

understanding of antimicrobial use. Given the primary healthcare setting of this audit, we adapted and piloted the GP version of the NAPS tool (GP NAPS) in remote clinics in northern WA, NT and Queensland.

Our aims were to:

1. Develop and pilot an audit tool of antimicrobial use for use in a remote primary healthcare setting

2. In piloting the tool across 15 primary healthcare clinics in three jurisdictions, we sought to describe:

   a. Similarities and differences in prescribing patterns by jurisdiction and healthcare professional status.
   b. Appropriateness of prescribing.
   c. Functionality of the audit tool in a qualitative sense

The following "Materials and Methods" section describes the process taken to achieve aim 1, and the "Results" section presents the descriptive analysis for aim 2.

## MATERIALS AND METHODS

### Adaption of the GP NAPS tool

Through an iterative process the study team, a multidisciplinary group of researchers, modified the exiting GP NAPS audit tool. Adapted data fields included additions that reflect the health professional types, antimicrobials and indications common in remote primary healthcare clinics, while removing fields that were not relevant. It was designed to provide consistent data interpretation across the different states and territories and by various auditor types (Appendix A).

The study team comprised:

1. Clinical pharmacists working within the jurisdictions,
2. National Centre for Antimicrobial Stewardship (NCAS) staff with experience in developing and using the GP NAPS audit tool,
3. Infectious diseases clinicians with expertise in antimicrobial stewardship or working in remote health,
4. Individuals with key oversight roles for remote health.

The study team provided relevant information on the differences between the various proposed primary healthcare clinics and possible requirements for additions or alterations to the GP NAPS audit tool. This required several rounds of template design and was conducted via email and teleconferences between 16/10/2017 and 26/03/2018.

Criteria used to assess appropriateness of antimicrobial use were identical to those used in the GP NAPS tool but tailored to local guidelines (Appendix B). In addition to the national Therapeutic Guidelines (eTG complete, 2015) (to which access and promotion of are requirements of the National Safety and Quality Health Service Standards (ACSQHC, 2017c) (NSQHS)), there are three regionally specific clinical guidelines used

across northern Australia: the Kimberley Clinical Protocols and Guidelines (KCPG) in northern WA (*KAHPF, 2016*), the Central Australian Rural Practitioners Association Standard Treatment Manual (CARPA) in the NT (*Remote Primary Health Care Manuals, 2017*) and the Primary Clinical Care Manual (PCCM) in Queensland (*Queensland Health, Royal Flying Doctor Service, 2019*). To assess variation in these guidelines, recommended antibiotic treatments for selected indications were compared (Table S1).

In this report we refer to antimicrobial 'use' instead of 'prescription' to reflect the range of health professionals who are legally able to supply and administer antimicrobials in the remote setting, specifically including doctors, remote area nurses and Aboriginal Health Practitioners.

## Selection of study sites

Five primary healthcare clinics of varying capacities were selected by local project leads in each of the three northern jurisdictions. These clinics were governed by the Kimberley Aboriginal Medical Services (KAMS) in WA, TEHS in the NT and Torres and Cape Hospital and Health Service (TCHHS) in Queensland.

## Audit process

Following remote training by NCAS, local auditors reviewed consecutive clinic presentations to identify the first 30 resulting in use of at least one antibiotic, antiviral, antifungal or antiparasitic agent. The same patient could be included more than once if treatment had not previously been given for the same indication.

Data were collected over a 2- to 3-week period via a standardized paper form (Appendix A) and later transferred to a Microsoft Excel (Office 365) spreadsheet. Auditors in northern WA and NT used remote access to electronic health records while in Queensland medical records were reviewed on site. All auditors were employees of the involved health services, familiar with local treatment guidelines.

Guideline compliance and appropriateness of each antimicrobial used was assessed against agreed definitions (Appendix B) (*NCAS and ACSQHC, 2018*), based on the NSQHS and Antimicrobial Stewardship Clinical Care Standard (*ACSQHC, 2018a*). In brief, antimicrobial use was "Appropriate" if it followed the local or Therapeutic Guidelines, or was endorsed by an infectious disease expert or clinical microbiologist, or was the most narrow-spectrum antimicrobial that covers the causative pathogen, or was a reasonable alternative to the guidelines (usually with justification for the choice documented in the notes). The alternative category, "Inappropriate", included antimicrobials used at the incorrect dose or too broad spectrum for the indication, or the choice was unlikely to treat the causative pathogen, or the indication didn't require antimicrobial treatment at all. The category "Not Assessable" was used if there was insufficient recorded information. The NAPS team at NCAS provided phone and email support to the local audit teams. Ten percent of antimicrobials audited in each facility were randomly reviewed by an expert at NCAS to assess reliability and validity of local scoring.

## Data analysis

The three jurisdictional datasets were transferred to a secure central repository and merged into a single dataset for analysis in Stata 15.1 (*StataCorp, 2017*). Descriptive analyses of the data were conducted and presented as proportions for categorical variables and medians for continuous data. No inferential statistical analyses were performed for this pilot study.

## Qualitative survey of end user functionality of the GP NAPS tool

A written questionnaire was used to survey local auditors to assess the practicality, usability, feasibility and generalisability of the audit tool to evaluate its suitability in context (Table S2). We did not aim to test the construct validity or reliability of the tool.

## Ethical approval

Ethical approval was obtained from each jurisdictional ethics committee (Western Australia Aboriginal Health Ethics Committee (WAAHEC): 839; Top End Health Research Ethics Committee (TEHREC): 2017–3012; Far North Queensland Human Research Ethics Committee (FNQ HREC): HREC/18/QCH/3, PHA RD007329, SSA/18/FNQ/9,10,11,12,13,14). A waiver of consent was granted because this research was retrospective, used de-identified data routinely collected at the time of healthcare provision and thus carried no risk to participants.

# RESULTS

## Cohort characteristics

A total of 668 antimicrobials for 529 patients between August 2017 and June 2018 were included (Table 1). The median age of patients was 23 years (interquartile range: 7–45 years), with 20% below 5 years. Females comprised 60% of patients. As expected, most patients were Aboriginal & Torres Strait Islander people.

## Trends in antimicrobial use

Nurses (305/668, 46%) and doctors (347/668, 52%) supplied/administered most of the antimicrobials with some variation across jurisdictions (Table 2). Aboriginal & Torres Strait Islander Health Practitioners rarely supplied/administered antimicrobials (4/668, <1%). This is generally reflective of the staff composition, although we might expect a higher proportion of Aboriginal Health Practitioners given that they comprise up to 10% of the workforce (*Zhao et al., 2017*; *Wright, Briscoe & Lovett, 2019*). Skin and soft tissue infections (SSTI) were the most common indication for antimicrobial use in all jurisdictions (WA: 35%; NT: 29%; QLD: 40%). The most frequently used antibiotics were benzathine benzylpenicillin and amoxicillin. Co-trimoxazole was more commonly used in Queensland than other jurisdictions (WA: 7%; NT: 5%; QLD: 18%).

Figure 2 reflects how different presentations are managed. In WA and the NT, SSTIs made up a slightly larger proportion of cases treated by nurses (42% and 32% respectively) compared with doctors (29% and 24% respectively). Furthermore, lower respiratory tract infections (LRTI) were treated by only doctors in WA, but by doctors and nurses in

**Table 1 Descriptive characteristics of audit sample population.**

|  | WA[A] | NT[A] | QLD[A] | Total |
|---|---|---|---|---|
| Audit dates | 02/03/2018– 02/06/2018 | 28/08/2017– 03/09/2017 | 10/05/2018– 15/06/2018 | 28/08/2017– 15/06/2018 |
| Unique patients | 186 (35%) | 162 (31%) | 181 (34%) | 529 |
| Antimicrobials used | 243 (36%) | 196 (29%) | 229 (34%) | 668 |
| Antimicrobials used with available microbiological result | 36 (15%) | 34 (17%) | 82 (36%) | 152 (23%) |
| Age, years (median [IQR[B]]) | 22 [5–44] | 24 [8–46] | 23 [8–44] | 23 [7–45] |
| Sex (M:F) | 39:61 | 33:67 | 49:51 | 40:60 |
| Aboriginal & Torres Strait Islander | 169 (91%; 8 unknown) | 153 (94%) | 141 (78%; 10 unknown) | 463 (88%; 18 unknown) |

Notes:
[A] WA, Kimberley region of Western Australia; NT, Top End of the Northern Territory; QLD, far north Queensland.
[B] IQR, Interquartile range.

**Table 2 Frequency (n (%)) of antimicrobial use by health professional, syndrome and antimicrobial.**

|  |  | WA[A] | NT[A] | QLD[A] | Total |
|---|---|---|---|---|---|
| Health professional | Doctors | 130 (53%) | 51 (26%) | 166 (72%) | 347 (52%) |
|  | Nurses | 100 (41%) | 142 (72%) | 63 (28%) | 305 (46%) |
|  | A&TSI[B] health practitioner | 1 (<1%) | 3 (2%) | 0 (0%) | 4 (1%) |
|  | Unknown | 12 (5%) | 0 (0%) | 0 (0%) | 12 (2%) |
|  | Total | 243 (100%) | 196 (100%) | 229 (100%) | 668 (100%) |
| Syndrome[C] | SSTI | 86 (35%) | 57 (29%) | 92 (40%) | 235 (35%) |
|  | GI | 28 (12%) | 19 (10%) | 19 (8%) | 66 (10%) |
|  | STI | 24 (10%) | 6 (3%) | 28 (12%) | 58 (9%) |
|  | LRTI | 20 (8%) | 35 (18%) | 8 (4%) | 63 (9%) |
|  | RHD | 6 (2%) | 22 (11%) | 14 (6%) | 42 (6%) |
|  | Other[D] | 79 (33%) | 57 (29%) | 68 (30%) | 204 (31%) |
|  | Total | 243 (100%) | 196 (100%) | 229 (100%) | 668 (100%) |
| Antimicrobial | Benzathine benzylpenicillin | 37 (15%) | 44 (22%) | 31 (14%) | 112 (17%) |
|  | Amoxicillin | 27 (11%) | 27 (14%) | 24 (10%) | 78 (12%) |
|  | Co-trimoxazole | 16 (7%) | 10 (5%) | 42 (18%) | 68 (10%) |
|  | Other[D] | 163 (67%) | 115 (59%) | 132 (58%) | 410 (61%) |
|  | Total | 243 (100%) | 196 (100%) | 229 (100%) | 668 (100%) |

Notes:
[A] WA, Kimberley region of Western Australia; NT, Top End of the Northern Territory; QLD, far north Queensland.
[B] A&TSI, Aboriginal and Torres Strait Islander.
[C] SSTI, skin and soft tissue infection; GI, gastrointestinal infection; STI, sexually transmitted infection; LRTI, lower respiratory tract infection; RHD, rheumatic heart disease.
[D] Other, all syndromes/antimicrobials with frequency <10% in every jurisdiction.

the NT. Secondary prophylaxis for rheumatic heart disease was used by only doctors in Queensland (14/14), by mostly nurses in the NT (18/22) and by doctors (3/6), nurses (2/6) and Aboriginal and Torres Strait Islander Health Practitioners (1/6) in WA (not shown in Fig. 2).

Peerⱼ

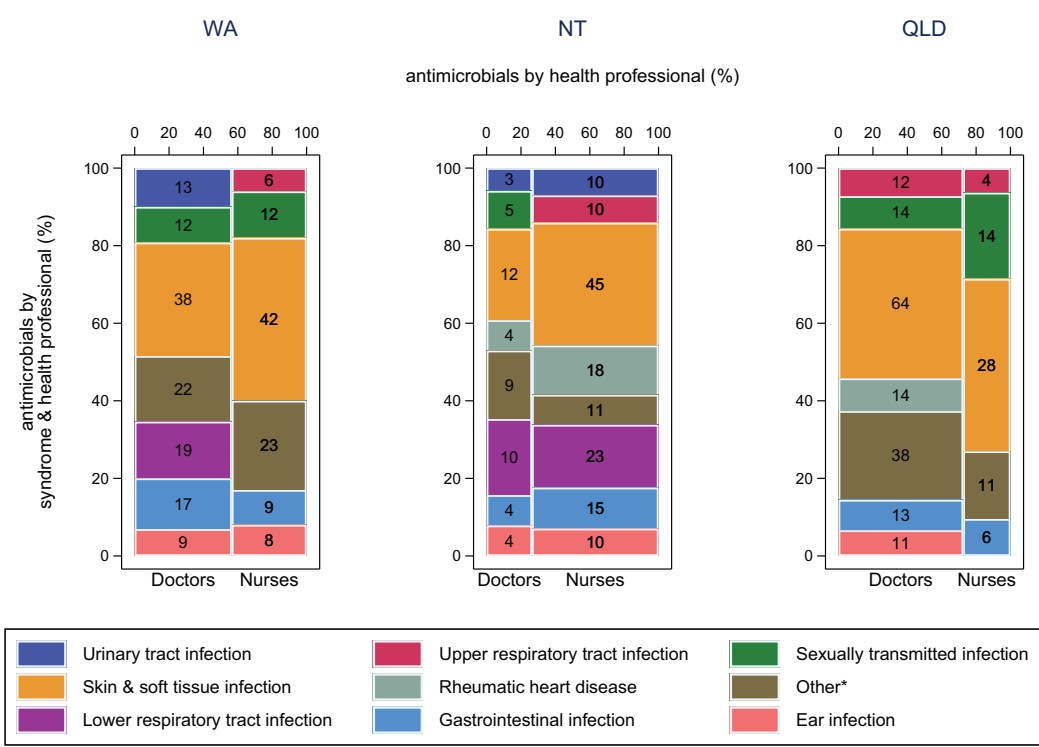

**Figure 2 Antimicrobial use by health professional and syndrome.** Frequency of antimicrobial use by health professional and syndrome in WA (Kimberley), the NT (Top End) and QLD (far north Queensland). Numbers in middle of columns are the number (not percentage) of antimicrobials used for that health professional/syndrome group. Total number of antimicrobials: WA–230, NT–193, QLD–229. *Other, all syndromes with frequency <5%.

For SSTIs (mainly impetigo and folliculitis/abscess) in WA and the NT, benzathine benzylpenicillin was more commonly used by nurses (43% and 29% respectively) than by doctors (24% and 17% respectively) (Fig. 3). Co-trimoxazole was used more in Queensland (43%) than in WA (16%) and the NT (14%), especially by doctors.

The second most common syndrome leading to antimicrobial use in the NT was LRTI (18%; WA: 8%, QLD: 4%). Remote Area Nurses treated most LRTIs in the NT (63%), using procaine penicillin (most often for pneumonia) or azithromycin (bronchiectasis) for 41% (Fig. S1). Sexually transmitted infections were common in both WA and Queensland (10% and 12% respectively; NT: 3%), with azithromycin frequently used (WA: 42%; QLD: 43%; NT: none), however there was notably more ceftriaxone used in Queensland (WA: 4%; QLD: 25%; NT: none) (Fig. S2).

## Appropriateness of antimicrobial use

More than 85% of antimicrobials used were considered appropriate in WA and the NT according to local or Therapeutic Guidelines (205/225 and 159/193 respectively) (Fig. 4). In Queensland, 65% (147/226) of antimicrobials used were considered appropriate. On average, 86% of antimicrobials used by nurses were appropriate (WA: 91% (90/99); NT: 87% (122/141); QLD: 77% (47/61)) (Table S3), compared with 73% of those used by doctors (WA: 90%; NT: 73%; QLD: 61%).

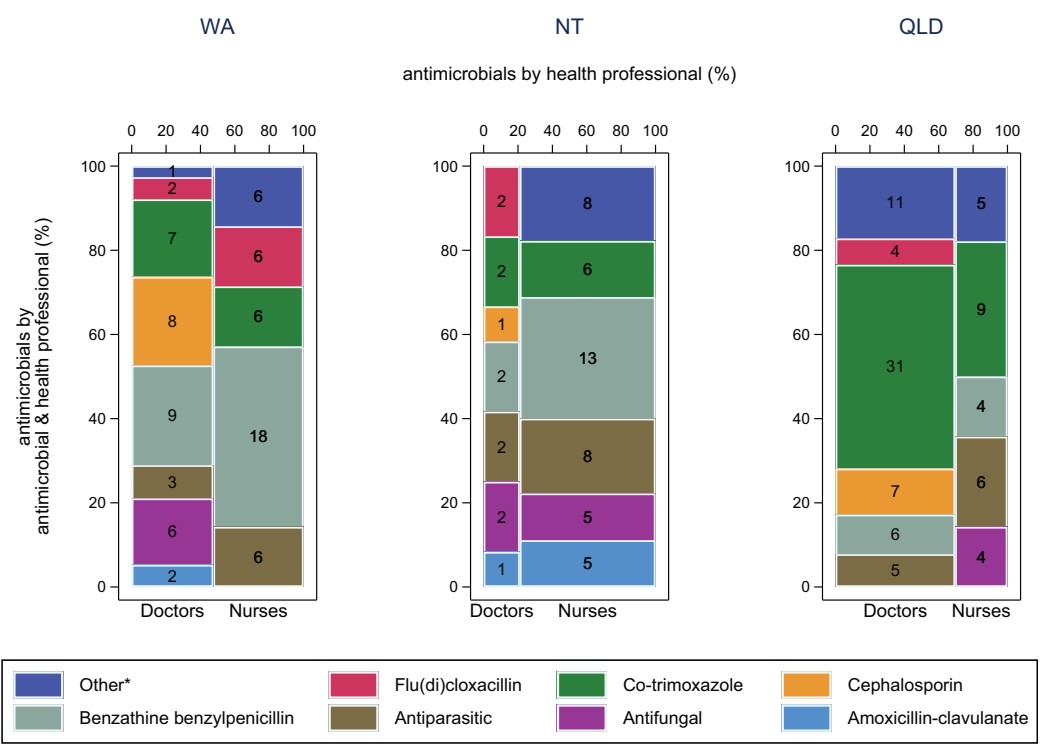

**Figure 3 Antimicrobial use by health professional and antimicrobial for skin and soft tissue infections.** Frequency of antimicrobial use by health professional and antimicrobial for skin and soft tissue infections in WA (Kimberley), NT (Top End) and QLD (far north Queensland). Numbers in middle of columns are the number (not percentage) of antimicrobials used for that health professional/ antimicrobial group. Total number of antimicrobials: WA–80, NT–57, QLD–92. *Other, all anti-microbials with frequency <5%.

In all jurisdictions, dose and frequency of antimicrobial treatment were the most commonly incorrect features of antimicrobial treatment (WA: 8%; NT: 7%; QLD: 13%) (Table S4). Other commonly incorrect features included antimicrobials used for an indication that did not require antimicrobial treatment (WA: 4%; NT: 7%; QLD: 10%), or with a duration that extended beyond the recommended period (WA: 7%; NT: 2%; QLD: 4%).

Antimicrobials used for trauma and non-surgical wound infections contributed substantially to inappropriate use by doctors in Queensland with 7/24 deemed appropriate (WA: 10/14; NT: 1/5). For wound infections, co-trimoxazole was used in 10 of 15 cases in Queensland (0/10 appropriate due to incorrect spectrum of activity and/or incorrect dosing when compared to the local or national guidelines) and in 4 of 7 in WA (4/4 appropriate). Overall, antimicrobials used for impetigo were mostly appropriate (WA: 17/18; NT: 9/10; QLD: 15/21) for which co-trimoxazole was used more often in Queensland (WA: 1/1 appropriate; NT: 2/2 appropriate; QLD: 9/10 appropriate) compared with benzathine benzylpenicillin in WA (16/16 appropriate) and the NT (7/8 appropriate).

Amoxicillin was often used inappropriately in Queensland by doctors for otitis media due to incorrect dose or frequency (WA: 3/3 (two nurses) appropriate; NT: 8/9 (all nurses)

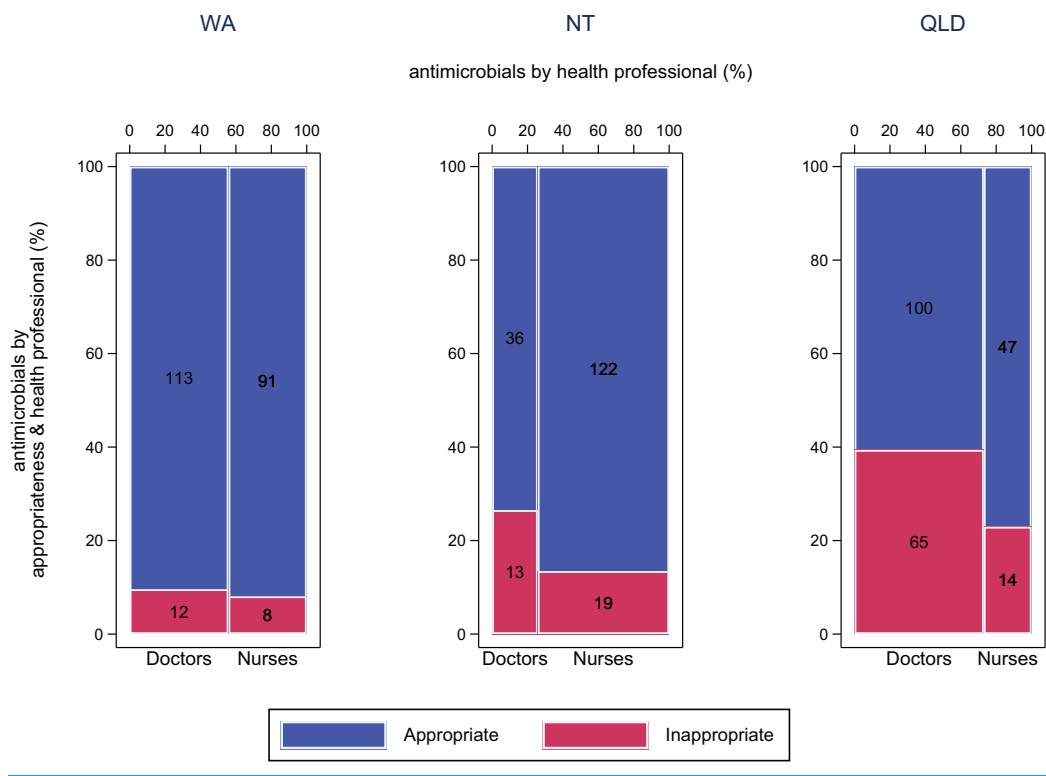

**Figure 4 Appropriateness of antimicrobial use by doctors and nurses.** Appropriateness of antimicrobial use by doctors and nurses, by jurisdiction WA (Kimberley), NT (Top End) and QLD (far north Queensland). Appropriate, Optimal or Adequate, Inappropriate, Suboptimal or Inadequate.

appropriate; QLD: 1/7 appropriate). Amoxicillin and amoxicillin-clavulanate were also used inappropriately for pneumonia in Queensland (WA: 5/5 appropriate; NT: 7/12 appropriate; QLD: 0/5 appropriate).

## Qualitative survey of end user functionality of the GP NAPS tool

While the tool was found to be acceptable in this context, auditors made some recommendations for improvement (Table S2), such as the addition of some indications common in the remote primary healthcare setting and training specific to the assessment of antimicrobials used in these cases to improve consistency across jurisdictions. An additional category to grade antimicrobials compliant with national guidelines but not with local guidelines was suggested as being useful for future analyses and to identify variations in guidelines. Likewise, stratifying by medical onsite and offsite (i.e., telephone order) review was suggested. Auditors reported a lack of detail in many clinical records, including absence of imaging, laboratory results and other patient medical history, which made identification of the indication and assessment of appropriateness difficult in some cases, especially at clinics that used only paper-based medical records. However, this is a challenge faced in any type of survey using administrative data, rather than a shortcoming of the tool itself, and is a common limitation of AMS in general (*Hawes et al., 2019*). The audit findings were reported back to each jurisdiction promptly which enabled

identification of action areas for their AMS programs. Auditors also highlighted areas in regional guidelines that could be refined to improve interpretation by less locally experienced staff and where guidelines could be updated to better align with the national Therapeutic Guidelines where appropriate.

## DISCUSSION

Our primary aim was to develop and pilot an audit tool of antimicrobial use for the remote primary healthcare setting in Australia. Adaptations made to an existing tool for urban primary healthcare use were relatively minor, incorporating health professionals, antimicrobials and indications common in remote primary healthcare as well as the Indigenous status of the patient. We gained useful preliminary insights on antimicrobial use, however these are descriptive findings that will require confirmation in future studies given that this audit included only a select few communities across northern Australia. Support from an expert team (at NCAS) was required to ensure consistency in data quality and interpretation. Overall the tool was suitable for use and was well received by the local health services in their first access to an audit tool.

The data demonstrated that SSTIs were the most frequent indications for antimicrobial use in all three regions (*Cuningham et al., 2019*; *Thomas et al., 2019*). This finding highlights the difference in morbidity profiles and patterns of antimicrobial use between remote and urban communities, as upper respiratory tract infections are typically the most common reason for an antimicrobial prescription in urban areas (*ACSQHC, 2019b*).

The appropriateness of antimicrobial use was high compared with urban GP settings around Australia. We found that narrow spectrum antibiotics like benzathine benzylpenicillin were frequently used in the remote setting, in contrast to the broader spectrum antibiotics like cephalexin and amoxicillin-clavulanate that are more often prescribed in urban primary healthcare (*Hawes et al., 2018*). Strong local ownership of region-specific guidelines combined with regulatory mechanisms to regulate the supply and provision of antimicrobials may have supported more standardized antimicrobial use. Furthermore, a higher rate of monthly benzathine benzylpenicillin injections for the treatment and prevention of rheumatic heart disease may increase overall appropriateness compared with other parts of Australia that have a proportionally smaller Indigenous population.

An important difference is that the health professionals supplying and administering antimicrobials differ in remote areas from urban settings. This study showed that nurses undertook a substantial proportion of this activity in remote areas whereas this is uncommon in primary care in urban areas. Antimicrobial use by Aboriginal & Torres Strait Islander Health Practitioners/Workers were lower than anticipated and further work is required to understand the barriers to antimicrobial use by this group (*Jongen et al., 2019*; *McCalman et al., 2019*; *Wakerman et al., 2019*).

Conflicts between multiple guidelines can influence antimicrobial use, and this issue can be exacerbated by high staff turnover rates in the remote setting (*Russell et al., 2017*). For example, at the time of the audit, the regional Queensland guidelines (PCCM) included co-trimoxazole for impetigo but not for folliculitis/abscess, while the KCPG

and CARPA include co-trimoxazole for both conditions. Similarly, co-trimoxazole for cellulitis was assessed as adequate treatment (i.e., in line with currently accepted medical practice) but non-compliant with the Therapeutic Guidelines, PCCM or other local guidelines. Some of these inconsistences have been resolved with release of the 10th edition of the PCCM in 2019 (Queensland Health, Royal Flying Doctor Service, 2019), which lists co-trimoxazole as indicated for impetigo and folliculitis/abscess, and for cellulitis in the Therapeutic Guidelines for people with an increased risk of MRSA infection (eTG complete, 2015). In future, electronic guidelines or decision support tools could be updated in a more timely manner.

### Next steps

This project aligns with the Australian Commission on Safety and Quality in Health Care's objectives of setting safety and quality goals for Aboriginal and Torres Strait Islander people in health service organizations, specifically agreeing on measures to analyze health outcomes and measuring performance towards targets (ACSQHC, 2017b; Posenelli et al., 2009). Embedding this research within existing services facilitates translation of research findings into ongoing practice and capacity building in AMS across remote clinics.

Changes to electronic software in the practice to improve indication documentation would make future auditing activity more efficient (Hawes et al., 2019). Automated data extraction and analysis would support remote AMS audits, improve assessment of antimicrobial use and understanding of the epidemiology of community-acquired infections, and could be linked to decision support tools. Real-time antimicrobial use data would complement other recent efforts in AMR surveillance across northern Australia (HOTspots: amrhotspots.com.au) and support clinical and public health efforts that aim to improve clinical care and reduce the emergence of AMR. A corresponding investment in workforce capacity in AMS is needed, across all relevant practitioner groups. Local Aboriginal engagement and leadership is essential in addressing AMR, the underlying burden of disease perpetuated by social determinants, and to prioritize the implementation of ongoing AMS in this setting. Furthermore, a focus on AMS in the accreditation standards of primary healthcare facilities is needed to enhance AMS activities and drive progress at the systems level.

## CONCLUSIONS

The development and piloting of an audit tool for antimicrobial use revealed that the morbidity profile and antimicrobial use in remote Aboriginal primary healthcare clearly differ compared with other primary care settings in Australia. The manual process of data collection was resource intensive. Broader implementation and scale up will likely require improved methods for automated data extraction.

## ACKNOWLEDGEMENTS

The authors acknowledge and thank the primary healthcare clinics and staff who participated in this project, as well as NCAS for their support. We also thank Ms Amy Legg for her role as Project Officer.

### Funding

This work was funded by a pilot project grant from the NHMRC-funded HOT NORTH program (GNT1131932). Will Cuningham is supported by an Australian Postgraduate Research Training Program Scholarship. Steven Tong is an Australian National Health and Medical Research Council Career Development Fellow (GNT1145033). Jodie McVernon is an Australian National Health and Medical Research Council Principal Research Fellow (GNT1117140). The funders had no role in study design, data collection and analysis, decision to publish, or preparation of the manuscript.

### Grant Disclosures

The following grant information was disclosed by the authors:
NHMRC-funded HOT NORTH program: GNT1131932.
Australian Postgraduate Research Training Program Scholarship.
Australian National Health and Medical Research Council Career Development Fellow: GNT1145033.
Australian National Health and Medical Research Council Principal Research Fellow: GNT1117140.

### Competing Interests

Steven Tong is an Academic Editor for PeerJ. Lorraine Anderson, Joanna Martin and Kerr Wright are employed by Kimberley Aboriginal Medical Services. Kathryn Daveson, Stacey McNamara and Trent Yarwood are employed by Queensland Statewide Antimicrobial Stewardship Program. Christine Connors, Bhavini Patel, and John Shanks are employed by Top End Health Service.

### Author Contributions

- Will Cuningham analyzed the data, prepared figures and/or tables, authored or reviewed drafts of the paper, and approved the final draft.
- Lorraine Anderson performed the experiments, authored or reviewed drafts of the paper, and approved the final draft.
- Asha C. Bowen conceived and designed the experiments, analyzed the data, authored or reviewed drafts of the paper, and approved the final draft.
- Kirsty Buising conceived and designed the experiments, performed the experiments, analyzed the data, authored or reviewed drafts of the paper, and approved the final draft.
- Christine Connors conceived and designed the experiments, analyzed the data, authored or reviewed drafts of the paper, and approved the final draft.
- Kathryn Daveson conceived and designed the experiments, analyzed the data, authored or reviewed drafts of the paper, and approved the final draft.
- Joanna Martin performed the experiments, authored or reviewed drafts of the paper, and approved the final draft.

- Stacey McNamara performed the experiments, analyzed the data, authored or reviewed drafts of the paper, and approved the final draft.
- Bhavini Patel conceived and designed the experiments, performed the experiments, analyzed the data, authored or reviewed drafts of the paper, and approved the final draft.
- Rodney James conceived and designed the experiments, performed the experiments, authored or reviewed drafts of the paper, and approved the final draft.
- John Shanks performed the experiments, analyzed the data, authored or reviewed drafts of the paper, and approved the final draft.
- Kerr Wright performed the experiments, analyzed the data, authored or reviewed drafts of the paper, and approved the final draft.
- Trent Yarwood conceived and designed the experiments, analyzed the data, authored or reviewed drafts of the paper, and approved the final draft.
- Steven YC Tong conceived and designed the experiments, analyzed the data, authored or reviewed drafts of the paper, and approved the final draft.
- Jodie McVernon conceived and designed the experiments, analyzed the data, authored or reviewed drafts of the paper, and approved the final draft.

### Human Ethics

The following information was supplied relating to ethical approvals (i.e., approving body and any reference numbers):

Ethical approval was obtained from each jurisdictional ethics committee (Western Australia Aboriginal Health Ethics Committee (WAAHEC): 839; Top End Health Research Ethics Committee (TEHREC): 2017-3012; Far North Queensland Human Research Ethics Committee (FNQ HREC): HREC/18/QCH/3, PHA RD007329, SSA/18/FNQ/9,10,11,12,13,14).

### Data Availability

The data is available in the Supplemental File.

### Supplemental Information

Supplemental information for this article can be found online at http://dx.doi.org/10.7717/peerj.9409#supplemental-information.

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
