# Peer review of "Antimicrobial stewardship in remote primary healthcare across northern Australia"

_PeerJ, doi:10.7717/peerj.9409_

## Round 0.1 · original submission · Minor Revisions

Please follow in details the Reviewer's recommendations.

·

Basic reporting

The authors sough to understand the antimicrobial use in remote primary healthcare settings in Northern Australia. This is an important study, given the rising burden of antimicrobial resistance, globally.

In general the paper is clearly written and easy to follow. However there are a few concerns regarding the basic reporting.
1. The use of the word "understanding" in the title is unnecessary as the authors did not measure this. This should be deleted
2. The authors inappropriately use certain words without definitions e.g. "episodes", clinical presentations, treating health professionals, broad spectrum and narrow spectrum etc. This is misleading. The authors should revise the manuscript and minimize the use of undefined terms or their misuse
3. The introduction, in particular does dont compare antimicrobial use with other studies/audits in Australia and outside Australia. There is need to do a thorough literature search to inform the need for the study
4. In general, it is not clear what the objectives of the study/audit were. Authors should state the primary objectives and secondary objectives. These seem to be avrying from the abstract, not indicated in the introduction, different in the methods, discussion and conclusions. There needs to be consistence.
5. The authors should clearly state the audit standards against which they were making comparisons. This is not mentioned any where!
6. There is duplication in reporting of results in the tables and figures. Kindly revise accordingly

Experimental design

(1) It is not clear what study design was implemented and the outome measures thereof. Although it is indicated as an audit (which is descriptive) the authors make several comparisons among the districts and other districts outside Northern Australia.
(2) The comparisons should be supported by valid statistical analysses such as ANOVA, t-test or Chi-square test or logistics regression
(3) results should be reported according to the objectives. gain if one objective is on the functionality of the tool, then appropriate tests should be done to test the construct validity and reliability of the tool in the context of Northern Australia.
(4) The authors should classify the antibiotics/antimicrobial according the agreed upon WHO/ATC classes and also quantify then according to WHO/ATC classes. This helps to copare results across countries and regions. Currently the even the pencilins are grouped differently and perhaps not well classified (e.g. Amoxicillin classified as a narrow spectrum antibiotic)
(5) The most challenging aspect of the manuscript is lack of clear objectives from the onset and throughout the sections
(6) There is need to define what remote is and describe these settings briefly. It is lao not clear why and how 5 clinics were chosen from each area and how many clinics are in each area. The inclusion exclusion criteria should be indicated, as well as the recruitment of patinets and how it was done!
(7) What was the software Stata used to analyse and why?

Validity of the findings

(1) The results should be present according to clearly defined objectives to guide clear reading of the manuscript

(2) The authors make alot of comparisons among districts, health professions, antimicrobial and regions outside Northern Australia which is not backed up by results or appropriate statistical analyses. Audits never aim to compare with others but to compare with self.

(3) The use of the tool in remote settings has to be tested for reliability and validity for the results to be interpreted accurately

(4) Who/how determined the appropriateness of the prescription. This may introduce bias and the authors need to explain this clearly

(4) The authors need to provide thresholds for all measurements , e.g. compliance, functionality of the tool, prescribing rates of antibiotics etc

·

Basic reporting

The authors should be congratulated for undertaking an ambitious body of auditing and analysis to provide a much needed insight into antibiotic prescribing in remote Australia. The following are recommendations for improving the article:

Background:
Given the international audience of this journal consideration should be given to providing more context regarding how the health system operates in Australia and in particular remote areas. This may also help with untangling all the acronyms.Many readers will be surprised to hear about how remote these areas are, and how clinical services are delivered as a result - including the staff mix and outreach services.
Some extra information on what we know about levels of antibiotic use and resistance in remote Australia from other studies and surveillance systems would also be useful.

Experimental design

This study was well designed and ethically conducted to a high technical standard.

Methods:
A quick summary of NAPS and why GP rather than hospital NAPS was chosen as a source for modification should be provided, along with relevant references.
Therapeutic Guidelines needs a reference and some explanation - esp in regards to how these are recommended by accreditation standards.
The evaluation of the tool is only allocated a few lines 146-148;215-224. In the discussion there is a focus in the paper on lack of documentation - but this really isn't an issue of the tool itself - more a challenge often faced in point prevalence surveys and AMS in general. There is a wealth of really interesting information and feedback about the tool hidden in the supplementary appendix. I suggest some of this is added into the main paper.

The presentation of results is clear, and in line with the stacked charts often used in this area. I found Figure 3 a bit confusing as it is trying to combine two sets of results - by jurisdiction, and also by profession. Consider splitting this into two parts or display like Figure 1 and 2 for consistency.

Validity of the findings

Discussion: Some of the most interesting findings in the table such as the % of patients who were Aboriginal & Torres Strait Islander, the difference in staffing profile between the jurisdictions, and the % of abx dispensed with micro results, are not discussed in enough detail (or at all). Were these results expected based on what we know about the settings? A bit of extra context would be useful here, rather than just a focus on what is different between urban and remote areas.

Additional comments

Well done in pulling this together - with just a bit of extra background and some more discussion it will make the article more accessible to an international audience.

---

## Round 0.2 · accepted · Accept

Thank you and congratulations!

·

Basic reporting

No comment

Experimental design

No comment

Validity of the findings

No comment

Additional comments

The authors have carefully edited the paper to address previous reviewer concerns. As a result the manuscript now reads very well, and provides important contextual information that is useful for readers not familiar with AMS in Australia.